This is a Registered Report and may have an associated publication; please check the article page on the journal site for any related articles.

REGISTERED REPORT PROTOCOL

# The effects of supervision on three different exercises modalities (supervised vs. home vs. supervised+home) in older adults: Randomized controlled trial protocol

**Sabrine Nayara Costa** [ID]*, **Luis Henrique Boiko Ferreira, Paulo Cesar Barauce Bento**

Physical Education Departament, Federal University of Paraná (UFPR), Curitiba, Brazil

* scosta713@gmail.com

## Abstract

### Background

Multicomponent physical exercise programs are a viable strategy for treating physical decline resulting from the aging process in older populations and can be applied in supervised and home-based modalities. However, the magnitude of the intervention effects in terms of physical function development may vary according to the modalities application due to different supervision degrees.

### Objective

This study aims to compare the effects of supervision in a multicomponent exercise program in different application modalities (supervised vs. home vs. supervised+home) in neuromuscular adaptations, muscle strength, gait, physical function, and quality of life, analyzing the differences between intensity, volume, and density of home and supervised sessions in community older adults.

### Methods

This protocol is a randomized controlled clinical trial with a sample of 66 older adults divided into three groups: supervised exercise (SUP = 22), home-based exercise (HB = 22), and supervised plus home-based exercise (SUP+HB = 22). The multicomponent exercise program will last 12 weeks, three times per week, for 60 min per session and include warm-up, balance, muscle-strengthening, gait, and flexibility exercises. The study's primary outcomes will be neuromuscular function, composed of the assessment of muscle isokinetic strength, muscle architecture, and neuromuscular electrical activation. The secondary outcome will be physical function, usual and maximum gait speed with and without dual-task, and quality of life. All outcomes will be assessed at baseline and post-intervention (week 12).

### Conclusion

This study will be the first clinical trial to examine the effects of different supervision levels on home-based exercises compared to supervised protocols. The results of this study will be essentials for planning coherent and viable home-based programs for older adults.

**Data Availability Statement:** Considering that this is a protocol study, the data will be provided in a full publication after the protocol acceptance.

**Funding:** The study was financed by the Universidade Federal do Paraná (UFPR) and Coordenação de Aperfeiçoamento de Pessoal de Nível Superior – Brasil (CAPES) – Finance Code 88887.373894/2019-00.

**Competing interests:** The authors have declared that no competing interests exist.

## Trial registration

Brazilian Registry of Clinical Trials. Number RBR- 7MZ2KR. https://apps.who.int/trialsearch/Trial2.aspx?TrialID=RBR-7mz2kr.

## Introduction

Physical exercise is recommended to prevent or reduce the effects of aging in older people and has been applied in the supervised and home-individual form [1–4]. In general, supervised programs are more effective than home-based programs for improving muscle strength, balance, physical function, and quality of life [3,5]. However, home-based programs are a good option for older adults with limited mobility or who need assistance to go to the training center and have little time for exercise due to family or work responsibilities.

In order to reduce the differences between home-based and supervised programs, recent studies have proposed adding a supervised-group session to home-based programs [6,7]. The addition of supervised-group sessions aims to provide greater exercise supervision, improving the control over frequency, intensity, and progression of exercises, which has been pointed out as one of the main limitations of home-based programs [8]. Nevertheless, control the effect of different supervision levels may directly affect the improvements in older adults' muscle strength and physical function. However, this hypothesis remains unclear due to the lack of information regarding the impact of different supervision levels in home-based programs.

The indirect supervision of home-based programs may result in limitations in the quality of exercise execution, resulting in exercises with a lower range of motion, less vigorous intensities, and longer rest interval between sets, impairing the exercise effects when compared to supervised training [8]. Thus, the strength gains with resistance exercise are generally associated with a combination of neural and morphological factors, such as increases in muscle mass, reductions in fat infiltration [9], improved firing rate, and decreased antagonist muscle co-activation (Mian., 2006) [10,11], which may not be so effective when performed at home.

To the best of our knowledge, this is the first study that aimed to analyze the neuromuscular mechanisms responsible for improvements in muscle strength and physical function in older adults with the addition of supervised sessions in home-based programs compared to totally unsupervised (home) and supervised training. Thus, we hypothesize that the addition of supervised sessions in home-based programs will lead to greater improvements in neuromuscular mechanisms due to the higher control over the training variables when compared to home-based programs. Besides, we believe that the addition of supervised sessions in home-based programs will provide similar neuromuscular benefits compared to fully supervised training interventions.

Therefore, understanding the influence of different exercise supervision levels on neuromuscular mechanisms and how it may change muscle strength and physical function seems essential for planning coherent exercise programs for older adults, expanding training strategies for that population. Thus, this study had two objectives: Compare the effects of a multicomponent exercise program in different application modalities (supervised vs. home vs. supervised+home) in neuromuscular adaptations, muscle strength, gait, physical function, and quality of life of older adults; and analyze the session intensity, volume, and density of different training modalities (supervised vs. home vs. supervised+home) s in community older adults.

## Materials and methods

### Trial design

It is a protocol for a randomized controlled clinical trial, investigator-blinded, with three parallel groups. Participants are individually randomized to one of the three groups in a 1:1:1 allocation ratio. The Research Ethics Committee approved the project at the Federal University of Paraná, Brazil (UFPR) (CAAE:20181219.4.0000.0102). The study protocol was developed based on the Standard Protocol Items: Recommendations for Interventional Trials (SPIRIT) guidelines [12]. The study follows the CONSORT guidelines [13], and all the items from the World Health Organization Trial Registration Data Set were registered in the publicly accessible Brazilian Registry of Clinical Trials (RBR-7MZ2KR). Any change made to the protocol will be reported to the Research Ethics Committee and modified in the Brazilian Registry of Clinical Trials.

The enrolled participants' assessments will be carried out at baseline prior to randomization, and after three months, as described below (Fig 1).

### Study setting

The study will be carried out in the city of Curitiba, Paraná, Brazil. In Paraná, the older population increased by 15.92% in the last five years, totaling 1.717.889 people [14]. In 2017, Curitiba registered 268.700 people aged over 60 years old. According to population projection assumptions, the number of older adults in Curitiba is expected to rise to 432.500 in 2030 and 544.500 in 2040.

### Eligibility criteria

The inclusion criteria will be: men and women aged > 60 years; be able to perform basic daily activities; not engaged in structured physical exercise program (previous six months); without cognitive impairment determined through the Mini-Mental State Examination [15].

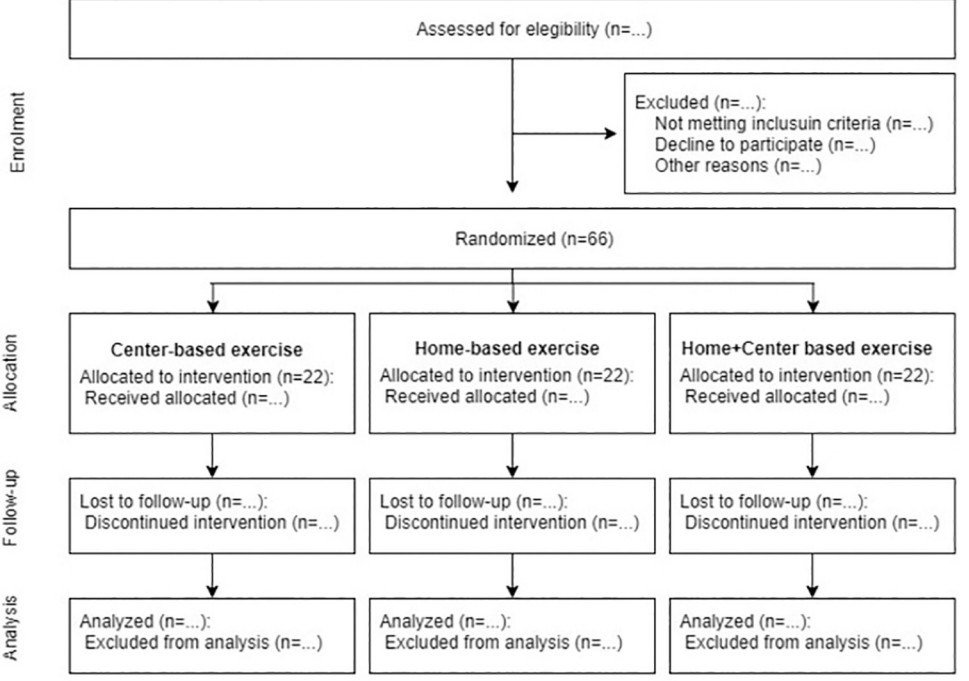

**Fig 1. Schematic representation of participant recruitment and allocation.**

Exclusion criteria include physical or motor limitations that make it impossible to perform functional tests; recent history of stroke or heart failure; uncontrolled acute or chronic metabolic disorders; severe osteoporosis; two or more fractures in the last year; medical contraindications for participation in an exercise program.

The participants will not be allowed to participate in another physical protocol concurrently with the present study.

## Recruitment, randomization, and treatment allocation

The study will be carried out in Curitiba, Paraná, Brazil. Volunteers will be recruited by dissemination through digital media (local digital newspaper and Facebook) and health units.

After checking the inclusion and exclusion criteria and analyzing the screening/anamnesis, the volunteers will sign a Free and Informed Consent Form. The participants will then be randomized by blocks into one of the three groups of equal size by a researcher who is not involved in the study. The randomization will be made at the randomization.com website. A second researcher not involved in the study will receive an envelope with the participants' allocation, contacting them to inform them about their specific training group. The researchers who carry out the evaluations and interventions will be blinded to the distribution of the groups.

## Intervention groups

Volunteers will participate in a multicomponent exercise program combining supervised and home-based exercise for 12 weeks, three times per week, 60 min per session. Participants will be divided into (1) supervised exercise (SUP), who will attend supervised group sessions at the training center; (2) home-based exercise (HB), who will participate in individual sessions at home; (3) supervised plus home-based exercise (SUP+HB), who will attend to one session per week at the training center and two sessions per week at home. Physical educators or physiotherapists will guide the supervised sessions at the training center.

The volunteers who will attend home sessions (HB and SUP+HB) will have an initial visit to their home to receive recommendations about how to perform the exercises safely. A guide with illustrations and instructions to reinforce proper exercise execution will be provided to each participant with space for recording the sessions performed and possible occurrences. On this occasion, they will receive ankle cuff weights (1, 2 and, 4 kg) to the strength training and a square stepping for walking exercises. To control program adherence, participants will record the days they complete the program, and they will also receive messages with incentives to perform the exercises at home.

The exercise program will consist of functional exercises of muscle strength, balance, flexibility, and gait, based on the recommendations of Singh et al. [16] and American College of Sports Medicine [17] (ACSM). Each session will be divided into 05 minutes of dynamic and articular warm-up, 10 minutes of dynamic and static balance exercises, 25 minutes of muscle-strengthening exercises, 10 minutes for gait exercises, and 10 minutes of flexibility exercises.

The exercise program will consist of four phases: phase 1 (1 to 3 weeks), phase 2 (4 to 6 weeks), phase 3 (7 to 9 weeks) and, phase 4 (10 to 12 weeks). Each phase will differ in the load and difficulty level of exercises, as shown in Table 1. The session intensity will be measured by the rate of perceived exertion (RPE) according to Borg Scale (6–20). In all groups analyzed, an explanation regarding the scale will be provided as well as a familiarization with the protocol before the start of interventions. Those measurements will be analyzed in different moments of the session, aiming to characterize the intensity perceived. In all training sessions, the volunteers will be encouraged to perform the exercises respecting the 1:3 ratio for strength exercises [18].

**Table 1. Multicomponent exercise program planning.**

| Modality | Balance Training | Resistance Training | Gait Training | Flexibility Training |
|---|---|---|---|---|
| Volume | 1–2 sets of 4–5 different exercises emphasizing static and dynamic postures for 30 seconds | **Phase 1:** 3 sets of 12 repetition, 1-min interval between sets<br>**Phase 2:** 3 sets of 10 repetition, 30-s interval between sets<br>**Phase 3:** 3 sets of 8 repetition, 1-min interval between sets<br>**Phase 4:** 3 sets of 8 repetition, 30-s interval between sets. | 1 set<br>5 repetition<br>6 gait exercises | Sustain stretching exercises for 20 sec to each major muscle groups |
| Intensity | Progressive difficulty with increases in complexity<br>**Phase 1:** reduction of foot support base.<br>**Phase 2:** unstable surface.<br>**Phase 3:** reduction of visual information.<br>**Phase 4:** objects manipulation. | **Phase 1:** 13–15 on the Borg Scale.<br>**Phase 2 to 4:** 15–17 on the Borg Scale. | **Phase 1 to 4:** 12–13 on the Borg Scale (40–60% of heart rate reserve) | **Phase 1 to 4:**<br>Progressive neuromuscular facilitation technique, which involves stretching as far as possible, then relaxing the involved muscles, then attempting to stretch further, and finally holding the maximal stretch position for at least 20 seconds. |
| Exercises | Static exercises: single-leg, semi-tandem, and tandem position, single-leg stand with knee elevation. Dynamic exercises: perform forward, backward, and lateral walking in plantarflexion, dorsiflexion, and tandem position. Transfers of chairs, move over objects, and step-ups. | **Phase 1 and 2:**<br>Sit-to-stand a chair; Standing knee; Pelvic lift; Wall sitting isometric; Sitting ankle plantar flexion; Standing ankle plantar flexion; Barbell curl; Bend-over dumbbell row; Halter front raise; Lying hip extension; Lying hip flexion; Sit-up; Walking lunges; Standing knee flexion; Standing hip extension; Hip adductor Exercise; Hip abductor Exercises<br>**Phase 3 and 4:**<br>Wall sitting isometric; Barbell curl; Bend-over dumbbell row; Sit-up; Walking lunges; Standing knee flexion; Squat; Seated Knee Extension; Pelvic lift; Standing hip flexion; Standing calf raise; Bent over row; Dumbbell curls; Dumbbell Side Lateral Raise; Hip extension; Stiff legged deadlift; Step-Ups; Abductor Exercises; Adductor Exercise. | Walk at the usual and maximum speed, and exercises involving square stepping. The square-stepping Exercise was based on the study conducted by Shigematsu colleagues (2008). | Exercises involving major muscle groups, including hip and knee flexion and extension, and abduction and adduction. Besides the addition of upper limb stretching movements. |

The training protocol will be modified if the participant undergoes severe pain or discomfort when carrying out the training program. Besides, the protocol will be interrupted if the participant is committed to any illness that may prevent exercise execution.

Aiming to enhance the validity of the presented data, several methods will be used to assess exercise protocol adherence, including engagement in the supervised exercise sessions, filling a calendar in the home sessions, and weekly calls to remember the importance of performing the exercise.

## Measurements

The evaluation sessions will be conducted in the biomechanics laboratory in CECOM at UFPR, where trained evaluators will administer the tests. The evaluators responsible for applying the exercise program will not participate in evaluating the sample, aiming to guarantee a simple blind study. The steps and timeline of the assessments can be observed in Fig 2.

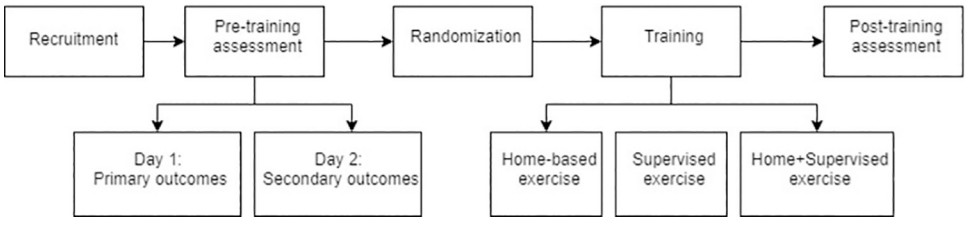

**Fig 2. Representation of the timeline and assessments of the study.**

All participants visited the laboratory on two nonconsecutive days with a minimum of 48 hours of resting to perform baseline evaluations. In the first visit, participants will be requested to answer a clinical anamnesis, followed by an anthropometric assessment. All protocols applied for primary and second outcomes will be described below.

## Primary outcomes

**Muscle isokinetic strength.** The evaluation of muscle isokinetic strength will be performed using the Biodex Multi-joint System dynamometer (Biodex Medical Systems. Inc. Shirley. NY. USA). The protocol includes two sets of three repetitions of the knee, hip, dorsiflexion, and plantarflexion concentric flexion/extension movements at 60˚/s and 180˚/s of dominant limb, with a two-minute interval between sets [19]. Participants will be positioned following the Biodex factory specifications, which are described by Symons et al. [20].

Each test will be performed twice at the same evaluation session, with different verbal commands of the evaluator: in the first test, a submaximal set will be used for familiarization, followed by a second test, which will be held with maximum strength for the record. Data will be acquired at 1000 Hz, and all muscle strength tests will be analyzed using the Biodex System 3 Advantage software, version 3.2. The variables analyzed will be peak torque/body weight, total work, and average power.

**Neuromuscular function.** The neuromuscular function will be evaluated using an isokinetic dynamometer Biodex along with surface electromyography (EMGs). Aiming to determine the voluntary activation level (VAL), the twitch interpolation technique will be used [21]. The present study protocol will stimulate the posterior tibial nerve, activating the sural triceps muscles (soleus and gastrocnemius) [22].

For the acquisition of EMG signals, an electromyograph (Trigno Wireless, Delsys, USA) will be used, with an electric stimulator (Neuro IOM-Neurosoft®) with bipolar surface electrodes AMBU Neuroline 715. The muscle analyzed will be the lateral gastrocnemius (LG) and medial gastrocnemius (MG) of the dominant limb, and the electrode placement process will follow the Surface EMG for the Non-Invasive Assessment of Muscles (SENIAM) recommendations [23,24]. Electrodes will be placed on the LG, MG, and tibialis anterior [23].

The reference electrode will be placed in a central position at the same leg of the stimulation, and an electrode for electrical stimulation will be placed over the patellar tendon. For the stimulation of the posterior tibial nerve, the cathode will be placed in the popliteal fossa.

After the electrodes are placed, the participant will be positioned on the isokinetic dynamometer to evaluate the ankle, as described by Symons et al. (2004). Data collection will be carried out during maximum voluntary isometric contraction (MVIC), which will consist of three maximal contractions of five seconds each.

The VAL will be analyzed through two supramaximal doublets (two electrical stimulation with 200 μs pulse width, 10 ms interstimulus interval, and 1000 mA maximum output) applied percutaneously femoral nerve during the MVIC. The optimal stimulation intensity will be

determined from the M wave and strength measurements before each test session. The stimulation intensity will increase by five mA until there is no further increase in the peak force contraction. Therefore, an addition of 30% of the highest value will be added to assure a supramaximal stimulation.

The first stimulation will be delivered during an MVIC, which will be nominated as the superimposed twitch. Followed the first stimulation, a second stimulus will be given 3 s after the MVIC, being nominated as the potentiated twitch. Both the STw and the PTw amplitudes allowed the VAL quantification, as proposed by Merton (1954). The variables analyzed will be the peak torque, contraction time, torque development rate analyzed through the formula (RTD = PT/CT), half-relaxation time, and VAL.

**Muscle architecture.**   The muscle architecture (fascicle length, pennation angle, and muscle thickness) of MG muscle of dominant limb will be assessed using an ultrasound (Konica Minolta Medical Imaging Inc Newark-Pompton Turnpike, Wayne, NJ, USA) in B mode with linear arrangement transducer (4 cm height x 2 cm length, 10 MHz), with the depth adjusted to 4 cm for the MG.

The image of MG will be performed with the participant resting in the prone position, with the feet hanging from the edge, and an ankle joint angle at 15˚ dorsiflexion will be fixed. The ultrasound transducer will be positioned at 30% of the portion between the fibula's lateral malleolus and the lateral condyle of the tibia [25]. The transducer will be oriented in the axial plane, aligned perpendicularly to the muscle, and moved from the center to the lateral position along with templates demarcated on the skin.

It is important to highlight that the gastrocnemius muscle is essential to maintain independence, gait speed and reduce falls risk. However, the aging process causes a muscle architecture alteration, reducing the cross-sectional area, pennation angle, and fascicle length. These age-related alterations resulting in a reduction of strength and muscle power, gait speed and functionality [26].

## Secondary outcomes

**Gait.**   The gait will be analyzed through the instrumented walking monitor ProtoKinetics Zeno Walkway (ProtoKinetics LLC, Havertown, Pennsylvania). The evaluation protocol will consist of four patterns: walking at the usual speed, maximum speed, the usual speed with a dual-task, and maximum speed with a dual-task for a distance of 10 meters. Dual-task gait will be assessed using an arithmetic cognitive task (countdown from 50) [27].

The parameters will be analyzed: gait speed (m/s), cadence (steps/min), stride time (secs), stride length (m), stride width (m), single and double support time (secs), and swing time (secs). Besides, the walk ratio (WR) will be obtained by dividing step length by cadence [28], and the locomotor rehabilitation index (LRI) is calculated as the percentage ratio between self-selected speed and optimum speed (algebraically LRI = 100 × self-selected speed/optimum walking speed) cc.

**Physical function.**   The physical function will be assessed using the Short Physical Performance Battery Test [29] and the Senior Fitness Test [30], aiming to verify muscle strength, agility, and dynamic and static balance. The Short Physical Performance Battery Test is composed of tandem, semi-tandem, side-by-side stands balance test, 4-meters walking speed test, and 5-times sit-to-chairs stand test. The Senior Fitness Test is composed of 5 times sit to chair stand test, arm curl test (number within 30 seconds), chair sit and reach test (distance between fingers and toe), back scratch test (distance between the two third fingers), Timed-up-and-go (time to rise, walk 3 m and return to the chair) and 6-minute walk test (distance walked in 6 minutes).

**Quality of life.** Quality of life will be assessed using the Medical Outcomes Study 36-item Short-Form Health Survey (SF-36) [31]. The SF-36 comprises 36 questions that cover eight health domains: physical limitation because of health problems, social limitation because of physical or emotional problem, usual role activities limitation because of physical health or emotional problems, body pain, general mental health, vitality, and general health perceptions.

**Physical activity level and exercise intensity control.** The physical activity level and the control of the exercise intensity will be evaluated using an accelerometer (Actigraph brand, model GT3X). To analyze the physical activity level, participants will be instructed to use the device for seven consecutive days, all day long, withdrawing only to sleep and to perform water activities, including bathing [32]. The accelerometer will be attached to the ankle of the dominant leg, just above the malleolus. It will be considered valid data the use of the accelerometer for at least four days, where at least one day has to be recorded during the weekend. The day will be considered valid when at least 10 hours are recorded. The data will be collected at a frequency of 60 Hz. The process of downloading and analyzing data will be carried out by the Actilife software using the Freedson equation [32].

The exercise intensity will be controlled by the use of accelerometers during the exercise sessions. The analyzed variable will be the average calories spent per day and session, sedentary time, light, moderate, vigorous, and very vigorous activities.

## Sample size

The sample size calculation was made using the G*Power software (version 3.1) [33]. A priori analysis was performed with the following input parameters: effect size (0.25) [7], type I error (0.05), type II error (0.95), number of groups (3), number of measurements (3), and a correlation between group (0.5). Also, the loss rate of 20% was considered. Thus, our analysis revealed a sample size of 66 subjects, divided into three groups of 22 participants.

## Adverse outcomes

Aiming to prevent the sample from performing exercises wrongly, and in order to avoid falls and fractures, the exercise will be monitored by physical education professionals as well as physiotherapists. Nevertheless, the participants will be monitored by the Rating of perceived Exertion to ensure the recommended intensity level, avoiding the deleterious effect of high-intensity exercise in older populations. In the case of muscle injury, physical training will be suspended.

## Statistical methods

For data analysis, the results will be reported using descriptive statistics and mean and standard deviation. All data will be analyzed by intention-to-treat, with baseline scores substituted for missing follow-up data. The Shapiro-Wilk test will be applied to verify the normality of the data and the Levene test for the sample's homogeneity. For parametric data, a comparison between groups and periods (pre and post-training) will be analyzed by an ANOVA of mixed-models with Bonferroni's post-hoc test. When differences between groups were observed at the baseline, a covariance analysis (ANCOVA) will be applied. In addition, a Pearson's correlation coefficient (r) was calculated to verify the magnitude of the possible observed effects, considering $r = .10$ as a small effect, $r = .30$ as a medium effect, and $r = .50$ as a large effect [34]. P-values less than 0.05 will be considered statistically significant. The tests will be performed using the IBM SPSS Statistics software version 25.

## Discussion

The present study's objective is to compare the effects of different supervision levels in a multi-component exercise program using three different application modalities in muscle strength, gait, physical function, and quality of life, verifying if different training modalities result in different neuromuscular adaptations. Finally, analyze if the addition of supervised sessions in home-based intervention promotes different impacts on the exercise intensity, volume, and density in community older adults.

We hypothesize that adding a supervised session in a home-based program will improve the neuromuscular function (muscle isokinetic strength, quality, and muscle composition), obtaining similar effects to a SUP exercise program, resulting in better effectiveness compared to a strictly HB intervention. It is already known that some factors as quality of exercise execution, range of motion, intensity, the interval between sets, and exercise progression may promote advantages in supervised training sessions compared to home-based interventions. Thus, we believe that the addition of a supervised session in the home-based group will provide direct feedback regarding the intensity, volume, and density of the physical exercise, promoting similar effects in the SUP+HB will be similar to SUP.

Our hypothesis is based on other studies that compared home-based programs with some degree of supervision. In a study conducted by Boshuizen et al. (2005), it was verified that maximal isometric knee strength and the walking speed significantly increase in the group with higher supervision (two group sessions supervised and one unsupervised home session per week) compared to the medium supervision group (one supervised group session and two unsupervised home sessions per week) [35]. Lacroix et al. (2016) found similar results regarding training-related improvements for the Romberg Test, stride velocity, Timed Up and Go Test, and Chair Stand Test in favor of the higher supervised group (two group sessions supervised and one unsupervised home session per week) compared to home-based program (three unsupervised home session per week). Therefore, it is possible to suggest that the addition of supervised sessions can enhance the effects of home-based exercises in older adults.

Thus, a systematic review [8] notices that the possible causes for SUP programs' superiority over HB interventions rely on improved control over the intensity, volume, and density of physical exercises made during the SUP program. Hence, it was verified through a meta-analysis that the SUP programs' superiority could be reduced or dismissed when small portions of supervision to HB programs were added. It suggests that the supervision made through calls, home visits, as well as attendance to training centers, would be an effective strategy to improve the exercise program's effectiveness.

However, to the best of our knowledge, the impact of improved supervision levels at HB interventions was not compared regarding the amount nor the type of supervision received during the exercise program in neuromuscular adaptations, isokinetic muscle strength, gait, physical function, and quality of life. Therefore, this is the first study that aims to verify if the addition of supervised sessions in a home-based program modifies the intensity, volume, and density of the exercise compared to SUP and HB intervention programs. Limitation of the study includes the spontaneous physical activity, such as the locomotion to the training site or daily activities (walk/transport), which will be monitored by accelerometry but will not be controlled in order to minimize or optimize the amount of time spent in these activities, since that type of activity must be performed in supervised group to go to the training site.

## Conclusion

This study aims to compare the impact of supervision on HB exercises to SUP protocols using a gold-standard method to verify the neuromuscular parameters, such as electromyography, to

analyze the muscle activation as well as the reaction time, elucidating the mechanisms underlying the muscle mass and strength development. The gold-standard method will provide a more consistent and accurate result, allowing a better understanding of those specific mechanisms. Knowing the effective contribution of neuromuscular mechanisms in muscle strength and physical function development with different exercise programs is essential for planning plausible and viable programs for older adults, expanding therapeutic strategies to older adults who have difficulty attending the training site.

## Supporting information

**S1 File.**
(PDF)

**S2 File.**
(PDF)

## Author Contributions

**Conceptualization:** Sabrine Nayara Costa, Luis Henrique Boiko Ferreira.

**Funding acquisition:** Sabrine Nayara Costa.

**Methodology:** Sabrine Nayara Costa, Luis Henrique Boiko Ferreira, Paulo Cesar Barauce Bento.

**Supervision:** Paulo Cesar Barauce Bento.

**Validation:** Sabrine Nayara Costa.

**Visualization:** Sabrine Nayara Costa, Luis Henrique Boiko Ferreira, Paulo Cesar Barauce Bento.

**Writing – original draft:** Sabrine Nayara Costa, Luis Henrique Boiko Ferreira.

**Writing – review & editing:** Sabrine Nayara Costa, Paulo Cesar Barauce Bento.

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
