## [Decision Letter · Decision Letter 0]

3 Nov 2020

PONE-D-20-25040

The effects of supervision on three different exercises modalities (center vs. home vs.  center+home) in older adults: Randomized controlled trial protocol

PLOS ONE

Dear Dr. Costa,

Thank you for submitting your manuscript to PLOS ONE. After careful consideration, we feel that it has merit but does not fully meet PLOS ONE’s publication criteria as it currently stands. Therefore, we invite you to submit a revised version of the manuscript that addresses the points raised during the review process.

The level of supervision in a multicomponent exercise program will be tested in older people. The study's question is timely and interesting to the audience of Plos One. While the reviewers and me found positive view on your paper, some major concerns were raised, particularly on reviewers 1 and 4. Please, reply all points carefully.

introduction

I think you can developing further your rationale for justifying the multicomponent program

lines 87-89 - Even interesting showing the basic rationale for neuromuscular adaptations, consider focusing on specific adaptations in elderly. For example, previous studies have shown that muscle weakness in aging leads to joint instability resulting in higher co-contraction levels and metabolic cost of walking (https://doi.org/10.1111/j.1748-1716.2006.01522.x and https://doi.org/10.1186/s40798-019-0228-6), possibly the multicomponent strategy can enhance primary outcomes of functionality as self-selected walking speed through neuromuscular and aerobic combined adaptations.

Another general suggestion is including two crucial markers of evaluation in gait analysis. the walking ratio (https://doi.org/10.1016/j.gaitpost.2018.08.030) and the locomotor rehabilitation index (10.4103/2468-5658.184750).

We look forward to receiving your revised manuscript.

Kind regards,

Leonardo A. Peyré-Tartaruga, Ph.D.

Academic Editor

PLOS ONE

Journal Requirements:

Reviewers' comments:

Reviewer's Responses to Questions

**Comments to the Author**

1. Does the manuscript provide a valid rationale for the proposed study, with clearly identified and justified research questions?

Reviewer #1: Partly

Reviewer #2: Yes

Reviewer #3: Yes

Reviewer #4: Partly

2. Is the protocol technically sound and planned in a manner that will lead to a meaningful outcome and allow testing the stated hypotheses?

Reviewer #1: Partly

Reviewer #2: Yes

Reviewer #3: Yes

Reviewer #4: Yes

3. Is the methodology feasible and described in sufficient detail to allow the work to be replicable?

Reviewer #1: Yes

Reviewer #2: Yes

Reviewer #3: No

Reviewer #4: Yes

4. Have the authors described where all data underlying the findings will be made available when the study is complete?

Reviewer #1: No

Reviewer #2: No

Reviewer #3: No

Reviewer #4: No

5. Is the manuscript presented in an intelligible fashion and written in standard English?

Reviewer #1: Yes

Reviewer #2: Yes

Reviewer #3: Yes

Reviewer #4: No

6. Review Comments to the Author

You may also provide optional suggestions and comments to authors that they might find helpful in planning their study.

Reviewer #1: General comments

This is an interesting randomized controlled trial protocol, with the objective of 'compare the effects of a multicomponent exercise program in different application modalities (center vs. home vs. center + home) in neuromuscular adaptations, muscle strength, gait, physical function, and quality of life 'and' analyze the differences between intensity, volume, and density of home and face-to-face sessions in community older adults'. It is an important starting point for evaluating different exercise programs for older adults.

The study is quite relevant and the findings will be of interest to professionals and scientists working with older populations. The manuscript is well and clearly written, the evaluation methods are adequate to answer the research question, requiring some adjustments.

Specific comments

Introduction:

- Updated referential, but some articles are with pre-frail elderly women, be careful with these studies.

- Consider inserting the stated hypothesis of the study (or primary outcomes).

- In general, the introduction is very well written, but I miss a specific focus on the research question, that is, addressing the subject of the two objectives of the work.

Methods:

- Consider registration with the International Clinical Trial Registry.

- I don't feel comfortable agreeing with the inclusion criterion (page 6 – 129) "not engaged in regular physical activity or exercise programs (previous six months)", I suggest analyzing this criterion.

- It is interesting to find other forms of dissemination for more people to be served (university page, health units, community center).

- Consider a screening / anamnesis of the patients together with the verification of the criteria, precious information appears at these moments.

- Figure 1 comments:

Many doubts arise regarding the use of the Borg Scale for the intensity of the exercises, especially for the group that will do the exercises at home.

Do the exercises seem to be focused only on the lower limbs? it is necessary to have a balance between lower and upper limbs, after all we also need them for the activities of daily living. Gait training also deserves more attention, proposing exercises at maximum walking speeds is interesting.

- Page 9 – 180: “…exercise. The participants will not be allowed to participate in another physical protocol concurrently with the present study.”

This stretch can be in the inclusion / exclusion criteria.

Measurements:

- The topic 'Measurements' can be accompanied by a figure explaining the steps and timeline of the assessment.

- 'Primary outcomes' are very well defined and explained, but 'Secondary outcomes' need to be explored (gait, Physical function and quality of life).

- Page 13 - 294 – “…and fractures, the exercise will be monitored individually by physical education professionals as well as physiotherapists.”

Reviewing this statement, 'individually' does not seem to be viable in this training model.

- Statistical methods - ANOVA will be used, I suggest using Generalized Estimation Equations (GHG), which is from the ANOVA family but more robust.

Discussion and Conclusion:

Page 20 - 313 to page 22 - 345:

- The hypothesis must be presented in the introduction and discussed here.

- The discussion is based only on one study (a systematic review) it is necessary to include others.

- It is important to emphasize the use of the gold-standard method, but it is necessary to include a paragraph that reflects on the benefits that the tested intervention will provide to the community and how it can be disseminated in the scientific community.

Reviewer #2: The proposed study will be a three-arm, randomized controlled clinical trial with 22 subjects in each arm to determine the effects of exercise supervision on three different exercise modalities. Outcomes and QOL will be assessed pre and post-intervention and compared between the three arms.

Minor revisions:

1- Line 285: Indicate the statistical testing method which achieves 95% power.

2- Line 309: Consider replacing the following sentence, “The coefficient of p <0.05 will be adopted to determine the significance.” with “P-values less than 0.05 will be considered statistically significant.”

3- Line 308: The magnitude of the coefficients appear to be more important than testing the null hypothesis that p=0. For this reason, possibly the p-values do not need to be reported.

Reviewer #3: The methodological and evaluation procedures are clearly written and are repeatable, however some points deserve attention:

- In the description of the proposed exercises it is not clear in sufficient detail to allow the work to be replicable especially at this stage. It is suggested to prepare a table with the phases (for weeks) as mentioned in the text, highlighting the phases of each session (warm-up, main part, calm down, etc ...) names of the exercises and the progression of volume and intensity.

- To assess muscle strength in the isokinetic dynamometer, it is recommended to evaluate the two lower limbs, so that it is possible to check the possible differences between the two segments in the pre and post intervention period and still make correlations if these values can interfere with balance static and dynamic subjects.

- It is not clear why the gastrocnemius muscles were chosen to assess muscle architecture.

- Why not evaluate the muscular architecture of the chosen muscles in the two lower limbs? it is likely that changes will occur after interventions that may in any way interfere with results such as strength and balance.

- The authors did not describe where all the data underlying the findings will be available when the study is completed.

- Report how the possible intervening variables of the study will be treated (uncontrolled) and how they can be minimized.

Reviewer #4: Dr. Sabrine Nayara Costa and co-authors present a registered protocol proposal entitled:The effects of supervision on three different exercises modalities (center vs. home vs. center+home) in older adults: Randomized controlled trial protocol. The study proposal addresses the interesting, albeit already very discussed topic of the effectiveness of home physical activity programs carried out without specialized supervision. The manuscript presents some grammatical inaccuracies about the conjugation of some verbs and the formation of some sentences. It should be corrected. Overall it appears interesting but requires significant changes.

First of all, I would not use the unclear term "center" to mean the supervised activity (commonly just referred as supervised). The abstract appears poor and not very specific in the backgroung part. Figure 1 is not complete and could be simplified or even removed and explained in the text.The “study setting” chapter presents unnecessary information. Hypertension or diabetes, very common pathologies in the over 60s, are not mentioned among the exclusion criteria.

In the introduction, the purpose of the research is not clear, in particular in the reasons for these measures and why these parameters should change following a different training mode (Muscle architecture, neuromuscular function ...). Better define the experimental question.

Line 153 sessions, not session

Line 170 does not seem correct to require maximum speed for all exercises, explain better.

The method of customizing the workload is real unclear. BORG scale? Explain in particular regarding resistance training. Was spontaneous physical activity not included in the workouts monitored all time long? For example, walking to go to the gym.

"Muscle strenght" should be defined as "Muscle isokinetic Strenght"

Line 319, Sentence not clear, better define the rationale for this hypothesis as already said several times

7. PLOS authors have the option to publish the peer review history of their article (what does this mean?). If published, this will include your full peer review and any attached files.

Reviewer #1: **Yes: **Valeria Feijo Martins

Reviewer #2: No

Reviewer #3: No

Reviewer #4: No

---

## [Author Response · Author response to Decision Letter 0]

14 Dec 2020

Dear Doctor Peyré-Tartaruga,

We would like to thank you for the opportunity to review and correct the previously submitted manuscript ID PONE-D-20-25040 - "The effects of supervision on three different exercises modalities (center vs. home vs. center+home) in older adults: Randomized controlled trial protocol". We appreciate the reviewers' comments and thank them for their thoughtful suggestions for improving our manuscript. We attended all the reviewer requests and included them in the document. Our point-by-point responses are detailed below.

ACADEMIC REVIEWER COMMENTS TO THE AUTHOR:

COMMENT 1:

Introduction

I think you can developing further your rationale for justifying the multicomponent program

lines 87-89 - Even interesting showing the basic rationale for neuromuscular adaptations, consider focusing on specific adaptations in elderly. For example, previous studies have shown that muscle weakness in aging leads to joint instability resulting in higher co-contraction levels and metabolic cost of walking (https://doi.org/10.1111/j.1748-1716.2006.01522.x and https://doi.org/10.1186/s40798-019-0228-6), possibly the multicomponent strategy can enhance primary outcomes of functionality as self-selected walking speed through neuromuscular and aerobic combined adaptations. 

Author Reply:

First of all, we would like to thank you for the suggestion. We have adjusted the introduction according to your comments in order to provide a better understanding of the manuscript objectives and rationality. We choose to use only one of the two recommended references for the introduction, considering the study objectives.

COMMENT 2:

Another general suggestion is including two crucial markers of evaluation in gait analysis. the walking ratio (https://doi.org/10.1016/j.gaitpost.2018.08.030) and the locomotor rehabilitation index (10.4103/2468-5658.184750).

Author Reply:

Thank you for the suggestions. Those analyses will contribute to a better understanding of the walking speed in older adults. Those analyses were added to the manuscript: 

Page 9, Line: 286 – 288:

“Besides, the walk ratio (WR) will be obtained by dividing step length by cadence 27, and the locomotor rehabilitation index (LRI) is calculated as the percentage ratio between self-selected speed and optimum speed (algebraically LRI = 100 × self-selected speed/optimum walking speed).”

 REVIEWER #1: 

General comments

This is an interesting randomized controlled trial protocol, with the objective of 'compare the effects of a multicomponent exercise program in different application modalities (center vs. home vs. center + home) in neuromuscular adaptations, muscle strength, gait, physical function, and quality of life 'and' analyze the differences between intensity, volume, and density of home and face-to-face sessions in community older adults'. It is an important starting point for evaluating different exercise programs for older adults.

The study is quite relevant and the findings will be of interest to professionals and scientists working with older populations. The manuscript is well and clearly written, the evaluation methods are adequate to answer the research question, requiring some adjustments.

Author reply: 

We want to thank you for all the comments. They were really interesting and will improve the manuscript quality. The answers to those comments will be presented in a point-by-point below.

COMMENT 1:

Introduction:

- Updated referential, but some articles are with pre-frail elderly women, be careful with these studies.

Author Reply:

Thank you for your observation regarding the references. However, we choose to keep those references since only two of them used pre-frail older adults. The first one (Garcia et al., 2020) was cited to demonstrate that home-based exercise is a viable strategy to prevent aging's deleterious effect. With that in mind, we also used studies with sarcopenic and dwelling community older adults. The other study used with pre-frail older adults was used to present strategies to improve control over home-based exercise. That study was cited along with a study with older adults with increased risk of falls. 

Considering that home-based exercise has been used as a tool for older adults with limited mobility or enhanced risk of adverse effects, it was necessary to use studies with specific older populations (sarcopenic, pre-frail, and fallers) to develop new researches in the field.

COMMENT 2:

- Consider inserting the stated hypothesis of the study (or primary outcomes).

- In general, the introduction is very well written, but I miss a specific focus on the research question, that is, addressing the subject of the two objectives of the work.

Author Reply:

Thank you for your suggestion. We observed that the introduction had some flaws regarding the hypothesis and research question, and therefore, we aimed to re-write the last paragraphs of the introduction. 

The modifications can be observed on page 4-5, line 90-104:

“The indirect supervision of home-based programs may result in limitations in the quality of exercise execution, resulting in exercises with a lower range of motion, less vigorous intensities, and longer rest interval between sets, impairing the exercise effects when compared to supervised training8. Thus, the strength gains with resistance exercise are generally associated with a combination of neural and morphological factors, such as increases in muscle mass, reductions in fat infiltration9, improved firing rate, and decreased antagonist muscle co-activation (Mian., 2006) 10,11, which may not be so effective when performed at home. 

To the best of our knowledge, this is the first study that aimed to analyze the neuromuscular mechanisms responsible for improvements in muscle strength and physical function in older adults with the addition of supervised sessions in home-based programs compared to totally unsupervised (home) and supervised training. Thus, we hypothesize that the addition of supervised sessions in home-based programs will lead to greater improvements in neuromuscular mechanisms due to the higher control over the training variables when compared to home-based programs. Besides, we believe that the addition of supervised sessions in home-based programs will provide similar neuromuscular benefits compared to fully supervised training interventions.”

COMMENT 3:

Methods:

- Consider registration with the International Clinical Trial Registry.

Author Reply:

Our study was registered and approved in the Brazilian Registry of Clinical Trials, associated with the International Clinical Trials Registry Platform: https://www.who.int/ictrp/network/rebec/en/

COMMENT 4:

- I don't feel comfortable agreeing with the inclusion criterion (page 6 – 129) "not engaged in regular physical activity or exercise programs (previous six months)", I suggest analyzing this criterion.

Author Reply:

We understand your concerns, and we choose to remove the regular physical activities from the inclusion and exclusion criteria. However, considering that the analyzed parameters (muscle strength, neuromuscular function, muscle architecture, gait, physical function, and quality of life) are influenced by structured physical activities, we choose to keep that parameter. Therefore, considering that this variable could be a confounder factor, the inclusion of physically active participants engaged in structured training programs could lead to misinterpretations.

Page 6, line 139:

“not engaged in structured physical exercise program (previous six months)”

COMMENT 5:

- It is interesting to find other forms of dissemination for more people to be served (university page, health units, community center).

Author Reply:

Thank you for the suggestion. The dissemination through digital media (local digital newspaper and Facebook) includes the University page and community center. However, we added the dissemination in health units. 

Page 7, line 149-150:

“The study will be carried out in Curitiba, Paraná, Brazil. Volunteers will be recruited by dissemination through digital media (local digital newspaper and Facebook) and health units.”

COMMENT 6:

- Consider a screening / anamnesis of the patients together with the verification of the criteria, precious information appears at these moments.

Author Reply:

Thank you for your suggestion; this information was added to the manuscript. 

Page 7, line 151:

“After checking the inclusion and exclusion criteria and analyzing the screening/anamnesis, the volunteers will sign a Free and Informed Consent Form.”

COMMENT 7:

- Figure 1 comments:

Many doubts arise regarding the use of the Borg Scale for the intensity of the exercises, especially for the group that will do the exercises at home.

Author Reply:

Thank you for your observation. However, Figure 1 is the schematic representation of participants' recruitment and allocation. Nevertheless, according to your comment regarding the Borg Scale, we believe that you are referring to Table 1. The session intensity will be measured by the rate of perceived exertion (RPE) according to Borg Scale (6-20). Those measurements will be analyzed in different moments of the session, aiming to characterize the intensity perceived. In all groups analyzed, an explanation regarding the scale will be provided as well as a familiarization with the protocol before the start of interventions. Concerning home-based intervention, the subjects will be instructed regarding the RPE protocol whenever necessary. That information was added in the manuscript on Page 8, Line 182-187. 

“The session intensity will be measured by the rate of perceived exertion (RPE) according to Borg Scale (6-20). In all groups analyzed, an explanation regarding the scale will be provided as well as a familiarization with the protocol before the start of interventions. Those measurements will be analyzed in different moments of the session, aiming to characterize the intensity perceived.”

COMMENT 8:

Do the exercises seem to be focused only on the lower limbs? it is necessary to have a balance between lower and upper limbs, after all we also need them for the activities of daily living. Gait training also deserves more attention, proposing exercises at maximum walking speeds is interesting.

Author Reply:

Thank you for your comment. Regarding the strength training, the exercise protocol will provide exercise for both, upper and lower body. The full description of the exercises can be observed in the updated Table 1, in exercises:

Page 8:

Phase 1 and 2: Sit-to-stand a chair; Standing knee; Pelvic lift; Wall sitting isometric; Sitting ankle plantar flexion; Standing ankle plantar flexion; Barbell curl; Bend-over dumbbell row; Halter front raise; Lying hip extension; Lying hip flexion; Sit-up; Walking lunges; Standing knee flexion; Standing hip extension; Hip adductor Exercise; Hip abductor Exercises

Phase 3 and 4: Wall sitting isometric; Barbell curl; Bend-over dumbbell row; Sit-up; Walking lunges; Standing knee flexion; Squat; Seated Knee Extension; Pelvic lift; Standing hip flexion; Standing calf raise; Bent over row; Dumbbell curls; Dumbbell Side Lateral Raise; Hip extension; Stiff legged deadlift; Step-Ups; Abductor Exercises; Adductor Exercise.

Regarding gait exercises, the subjects will perform several exercises using different speeds and with different stimulus. The information about gait exercises are provided on page 8, Table 1, exercise section: 

“Gait: Walk at the usual and maximum speed, and exercises involving square stepping. The square-stepping Exercise was based on the study conducted by Shigematsu colleagues (2008).”

COMMENT 9:

- Page 9 – 180: “…exercise. The participants will not be allowed to participate in another physical protocol concurrently with the present study.”

This stretch can be in the inclusion / exclusion criteria.

Author Reply:

Thank you for your suggestion. The phrase was reallocated to the Eligibility criteria section Page Line. 

Page 7, line 145-146:

“The participants will not be allowed to participate in another physical protocol concurrently with the present study.”

COMMENT 10:

Measurements:

- The topic 'Measurements' can be accompanied by a figure explaining the steps and timeline of the assessment.

Author Reply:

Thank you for the suggestion, and the figure was added in the Page 10, Line 207. 

COMMENT 11:

- 'Primary outcomes' are very well defined and explained, but 'Secondary outcomes' need to be explored (gait, Physical function and quality of life).

Author Reply:

Thank you for your observation. Due to a limited number of words, we choose to precisely describe the primary outcomes, considering that several different methods and protocols can be used to test the purposed outcomes. The battery test and the questionary of the secondary outcomes (Short Physical Performance Battery Test, Senior Fitness Test, and SF-36) are standard and widely described in the literature. However, considering your observation, we added the description of the secondary outcomes at Page 14, Line 293-305.

“Physical function: 

The Short Physical Performance Battery Test is composed of tandem, semi-tandem, and side-by-side stands balance test, 4-meters walking speed test, and 5-times sit to chair stand test. The Senior Fitness Test is composed of 5 times sit to chair stand test, arm curl test (number within 30 seconds), chair sit and reach test (distance between fingers and toe), back scratch test (distance between the two third fingers), Timed-up-and-go (time to rise, walk 3 m and return to the chair) and 6-minute walk test (distance walked in 6 minutes).

Quality of life: 

The SF-36 comprises 36 questions that cover eight health domains: physical limitation because of health problems, social limitation because of physical or emotional problem, usual role activities limitation because of physical health or emotional problems, body pain, general mental health, vitality, and general health perceptions.”

COMMENT 12:

- Page 13 - 294 – “…and fractures, the exercise will be monitored individually by physical education professionals as well as physiotherapists.”

Reviewing this statement, 'individually' does not seem to be viable in this training model.

Author Reply:

The word individually used in the mentioned phrase referred to the contact with the older adult through phone-calls or home-based visits, not necessarily to control supervised group sessions. However, we understand that the word "individually" could lead to misunderstandings, suggesting an individual control of the exercise session; therefore, we aimed to remove the word "individually" from the text, which did not affect the phrase context.

Page 15, line 330-331:

“…the exercise will be monitored by physical education professionals as well as physiotherapists.”

COMMENT 13:

- Statistical methods - ANOVA will be used, I suggest using Generalized Estimation Equations (GHG), which is from the ANOVA family but more robust.

Author Reply:

Thank you for your suggestion. We choose to use this analysis because the study objective is usually analyzed by an ANOVA mixed models, allowing the evaluation of the effects of different exercise programs across interventions (within groups) as well as to compare the effect between groups. Although we are not changing it in the present manuscript, we will consider this analysis method (GHG) during the data analysis after the protocol application. 

COMMENT 14:

Discussion and Conclusion:

Page 20 - 313 to page 22 - 345:

- The hypothesis must be presented in the introduction and discussed here.

- The discussion is based only on one study (a systematic review) it is necessary to include others.

- It is important to emphasize the use of the gold-standard method, but it is necessary to include a paragraph that reflects on the benefits that the tested intervention will provide to the community and how it can be disseminated in the scientific community.

Author Reply:

Thank you for your suggestions. We have reviewed the Discussion and Conclusion in order to improve it according to your request. The modifications can be observed in Page 16-18. 

REVIEWER #2: 

The proposed study will be a three-arm, randomized controlled clinical trial with 22 subjects in each arm to determine the effects of exercise supervision on three different exercise modalities. Outcomes and QOL will be assessed pre and post-intervention and compared between the three arms.

COMMENT 1:

Minor revisions:

1- Line 285: Indicate the statistical testing method which achieves 95% power.

Author Reply: 

We indicated that the power achieved is 95% in the following sentence: 

Page 15, line 322-326:

“A priori analysis was performed with the following input parameters: effect size (0.25) 7, type I error (0.05), type II error (0.95), number of groups (3), number of measurements (3), and a correlation between group (0.5).” 

Thus, the Alpha (type I error) used was 0.05, and Beta (type II error) used was 0.95, providing the desired power level (95%). 

COMMENT 2:

2- Line 309: Consider replacing the following sentence, “The coefficient of p <0.05 will be adopted to determine the significance.” with “P-values less than 0.05 will be considered statistically significant.”

Author Reply:

Thank you for your suggestion. The phrase was adjusted according to your request.

Page 16, line 344-345:

“P-values less than 0.05 will be considered statistically significant.”

COMMENT 3:

3- Line 308: The magnitude of the coefficients appear to be more important than testing the null hypothesis that p=0. For this reason, possibly the p-values do not need to be reported.

Author Reply:

Thank you for your suggestion. We agree that the magnitude of the coefficient is more appropriate to define the impact of the training method in older adults since small changes can be observed in the magnitude of the effect. Therefore, we will use your suggestion during the data analysis, and we removed the idea of P's coefficient as reported above.

Reviewer #3: 

The methodological and evaluation procedures are clearly written and are repeatable, however some points deserve attention:

COMMENT 1:

- In the description of the proposed exercises it is not clear in sufficient detail to allow the work to be replicable especially at this stage. It is suggested to prepare a table with the phases (for weeks) as mentioned in the text, highlighting the phases of each session (warm-up, main part, calm down, etc ...) names of the exercises and the progression of volume and intensity.

Author Reply:

Thank you for your suggestion. We changed the Table 1 in order to provide a better view of the exercise program. The modifications can be observed in page 9, line 191.

COMMENT 2:

- To assess muscle strength in the isokinetic dynamometer, it is recommended to evaluate the two lower limbs, so that it is possible to check the possible differences between the two segments in the pre and post intervention period and still make correlations if these values can interfere with balance static and dynamic subjects.

Author Reply:

We understand the importance of evaluating both lower limbs in an isokinetic dynamometer, and we already used that method in our research group in other studies. However, in the present study, we had some reasons to choose the analysis in only one limb; the first one is related to the study objective, which does not include the evaluation of the difference between limbs and dynamic/static balance after the training intervention. Besides, the post-intervention evaluation of the same limb will provide the answers related to the neuromuscular system and the training method.; the second is related to Biodex complex setups that include the calibration of some equipment (EMG and Neuro IOM-Neurosoft®) for neuromuscular evaluations. The third reason is related to the time spent during the evaluations. The protocol described in the present study last at least 4 hours (divided into two days) to be completed, so the addition of another limb would promote a significant increase in the time spent during the analysis. In previous studies conducted in our lab, we lost several individuals during our analysis process due to the long time spent during the initial assessments. 

COMMENT 3:

- It is not clear why the gastrocnemius muscles were chosen to assess muscle architecture.

Author Reply:

Thank you for the suggestion. The information was added to the manuscript.

Page 13, line 296-299:

It is important to highlight that the gastrocnemius muscle is essential to maintain independence, gait speed and reduce falls risk. However, the aging process causes a muscle architecture alteration, reducing the cross-sectional area, pennation angle, and fascicle length. These age-related alterations resulting in a reduction of strength and muscle power, gait speed and functionality 26. 

COMMENT 4:

- Why not evaluate the muscular architecture of the chosen muscles in the two lower limbs? it is likely that changes will occur after interventions that may in any way interfere with results such as strength and balance.

 Author Reply:

We understand the importance of evaluating both lower limbs. However, in the present study, we had two reasons to choose the analysis in only one limb; the first one is related to the study objective, which does not include evaluating the difference between limbs; the second is associated with the time spent during the evaluations. The protocol described in the present study last at least 4 hours (divided into two days) to be completed, so the addition of another limb (in both, isokinetic dynamometer and muscle architecture) would promote a significant increase in the time spent during the analysis. 

Nevertheless, the biggest part of studies that aimed to analyze the effect of training interventions in muscle architecture only analyzed the dominant limb. Thus, since it was more viable considering the problems mentioned above, only the dominant limb will be analyzed. 

COMMENT 5:

- The authors did not describe where all the data underlying the findings will be available when the study is completed.

Author Reply:

Thank you for pointing that out. That information was added in Page 13, Line 114-115:

“Data Availability: All relevant data are within the paper and its Supporting Information files.”

COMMENT 6:

- Report how the possible intervening variables of the study will be treated (uncontrolled) and how they can be minimized.

Author Reply:

Thank you for your suggestion. The spontaneous physical activity, such as the locomotion to the training site or daily activities (walk/transport), will be monitored by accelerometry but will not be controlled in order to minimize or optimize the amount of time spent in these activities, since that type of activity must be performed in supervised group to go to the training site. That information will be added at the end of the discussion as a limitation to the present study Line 18, Page 387-391:

“Limitation of the study includes the spontaneous physical activity, such as the locomotion to the training site or daily activities (walk/transport), which will be monitored by accelerometry but will not be controlled in order to minimize or optimize the amount of time spent in these activities, since that type of activity must be performed in supervised group to go to the training site.”

REVIEWER #4:

 Dr. Sabrine Nayara Costa and co-authors present a registered protocol proposal entitled:The effects of supervision on three different exercises modalities (center vs. home vs. center+home) in older adults: Randomized controlled trial protocol. The study proposal addresses the interesting, albeit already very discussed topic of the effectiveness of home physical activity programs carried out without specialized supervision. The manuscript presents some grammatical inaccuracies about the conjugation of some verbs and the formation of some sentences. It should be corrected. Overall it appears interesting but requires significant changes.

First of all, I would not use the unclear term "center" to mean the supervised activity (commonly just referred as supervised). 

The abstract appears poor and not very specific in the backgroung part. 

Figure 1 is not complete and could be simplified or even removed and explained in the text.

The “study setting” chapter presents unnecessary information. Hypertension or diabetes, very common pathologies in the over 60s, are not mentioned among the exclusion criteria.

In the introduction, the purpose of the research is not clear, in particular in the reasons for these measures and why these parameters should change following a different training mode (Muscle architecture, neuromuscular function ...). Better define the experimental question.

Author Reply:

Thank you for your suggestions. Your suggestions were quite interesting and will improve the quality of the manuscript. We will answer all the questions and appointments made in your revision in a point-by-point that can be observed below. 

COMMENT 1:

The manuscript presents some grammatical inaccuracies about the conjugation of some verbs and the formation of some sentences. It should be corrected.

Author Reply:

The manuscript underwent a complete grammatical revision to adjust some verbs' conjugation and the formation of some sentences according to the proper English. 

COMMENT 2:

First of all, I would not use the unclear term "center" to mean the supervised activity (commonly just referred as supervised). 

Author Reply:

 Thank you for your observation. The term center was changed to supervised exercise. 

COMMENT 3:

The abstract appears poor and not very specific in the backgroung part. 

Author Reply:

Thank you for recommending an adjustment in the abstract. The changes are in the abstract to adjust it according to our hypothesis. 

COMMENT 4:

Figure 1 is not complete and could be simplified or even removed and explained in the text.

Author Reply:

Figure 1 was added according to SPIRIT 2013 Statement recommendations: Defining Standard Protocol Items for Clinical Trials. More specifically, Figure 1 answer the item “Participant timeline” - Time schedule of enrollment, interventions (including any run-ins and washouts), assessments, and visits for participants. A schematic diagram is highly recommended (Figure). Besides, PLoS One requests a checklist of all SPIRIT items; therefore, we choose to keep Figure 1.

COMMENT 5:

The “study setting” chapter presents unnecessary information.

Author Reply:

The “study setting” was added according to SPIRIT 2013 Statement recommendations: Defining Standard Protocol Items for Clinical Trials. PLoS One requests a checklist of all SPIRIT items; therefore, we choose to keep the study setting. 

COMMENT 6:

Hypertension or diabetes, very common pathologies in the over 60s, are not mentioned among the exclusion criteria.

Author Reply:

Thank you for your suggestion. Nevertheless, we consider that this information is informed when we cite “uncontrolled acute or chronic metabolic disorders” in Eligibility Criteria. We only choose to exclude the older adults that present those pathologies without the proper control, considering that 60% of the Brazilian older population have hypertension, and 25% have diabetes.

COMMENT 7:

In the introduction, the purpose of the research is not clear, in particular in the reasons for these measures and why these parameters should change following a different training mode (Muscle architecture, neuromuscular function ...). Better define the experimental question.

Author Reply:

Thank you for your suggestion. We observed that the introduction had some flaws regarding the hypothesis and research question, and therefore, we aimed to re-write the last paragraphs of the introduction. The modifications can be observed on page 4-5, line 90-104:

“The indirect supervision of home-based programs may result in limitations in the quality of exercise execution, resulting in exercises with a lower range of motion, less vigorous intensities, and longer rest interval between sets, impairing the exercise effects when compared to supervised training8. Thus, the strength gains with resistance exercise are generally associated with a combination of neural and morphological factors, such as increases in muscle mass, reductions in fat infiltration9, improved firing rate, and decreased antagonist muscle co-activation (Mian., 2006) 10,11, which may not be so effective when performed at home. 

To the best of our knowledge, this is the first study that aimed to analyze the neuromuscular mechanisms responsible for improvements in muscle strength and physical function in older adults with the addition of supervised sessions in home-based programs compared to totally unsupervised (home) and supervised training. Thus, we hypothesize that the addition of supervised sessions in home-based programs will lead to greater improvements in neuromuscular mechanisms due to the higher control over the training variables when compared to home-based programs. Besides, we believe that the addition of supervised sessions in home-based programs will provide similar neuromuscular benefits compared to fully supervised training interventions.”

COMMENT 8:

Line 153 sessions, not session

Author Reply:

Thank you for the correction. The word was correct as recommended. 

COMMENT 9:

Line 170 does not seem correct to require maximum speed for all exercises, explain better.

Author Reply:

Thank you for your correction. The idea of “as fast as possible” was removed from the manuscript since the contraction time should respect the 1:3 ratio for strength exercises Page 8, Line 186-187:

“In all training sessions, the volunteers will be encouraged to perform the exercises respecting the 1:3 ratio for strength exercises.”

COMMENT 10:

The method of customizing the workload is real unclear. BORG scale? Explain in particular regarding resistance training. Was spontaneous physical activity not included in the workouts monitored all time long? For example, walking to go to the gym.

Author Reply:

Adjustments in the Figure 1 (page 9, line 191) were made to provide a better understanding of the workload used during at each phase of the intervention.

The session intensity will be measured by the rate of perceived exertion (RPE) according to Borg Scale (6-20). Those measurements will be analyzed in different moments of the session, aiming to characterize the intensity perceived. In all groups analyzed, an explanation regarding the scale will be provided as well as a familiarization with the protocol before the start of interventions. This information was add in page 8, line 182-186.

The spontaneous physical activity, such as the locomotion to the training site or daily activities (walk/transport), will be monitored by accelerometry but will not be controlled in order to minimize or optimize the amount of time spent in these activities, since that type of activity must be performed in supervised group to go to the training site. That information will be added at the end of the discussion as a limitation to the present study Line 18, Page 387-391. Nevertheless, since we will have the accelerometry data, a further analysis regarding the time spent in activities can be performed to verify if the home-based approaches promote different changes in the physical habits. 

COMMENT 11:

"Muscle strenght" should be defined as "Muscle isokinetic Strenght"

Author Reply:

Thank you for your suggestion. The term “Muscle Strength” was replaced by “Muscle Isokinetic Strength” when appropriated. 

COMMENT 12:

Line 319, Sentence not clear, better define the rationale for this hypothesis as already said several times

Author Reply:

Thank you for your recommendation. The whole discussion was adjusted to provide a better understanding of our hypothesis. Page 16-18.

---

## [Decision Letter · Decision Letter 1]

28 Oct 2021

The effects of supervision on three different exercises modalities (supervised vs. home vs.  supervised+home) in older adults: Randomized controlled trial protocol

PONE-D-20-25040R1

Dear Dr. Costa,

We’re pleased to inform you that your manuscript has been judged scientifically suitable for publication and will be formally accepted for publication once it meets all outstanding technical requirements.

Kind regards,

John W. Apolzan, PhD

Academic Editor

PLOS ONE

Additional Editor Comments (optional):

Reviewers' comments:

Reviewer's Responses to Questions

**Comments to the Author**

1. Does the manuscript provide a valid rationale for the proposed study, with clearly identified and justified research questions?

Reviewer #1: Yes

Reviewer #2: Yes

Reviewer #3: Yes

2. Is the protocol technically sound and planned in a manner that will lead to a meaningful outcome and allow testing the stated hypotheses?

Reviewer #1: Yes

Reviewer #2: Yes

Reviewer #3: Yes

3. Is the methodology feasible and described in sufficient detail to allow the work to be replicable?

Reviewer #1: Yes

Reviewer #2: Yes

Reviewer #3: Yes

4. Have the authors described where all data underlying the findings will be made available when the study is complete?

Reviewer #1: Yes

Reviewer #2: Yes

Reviewer #3: Yes

5. Is the manuscript presented in an intelligible fashion and written in standard English?

Reviewer #1: Yes

Reviewer #2: Yes

Reviewer #3: Yes

6. Review Comments to the Author

You may also provide optional suggestions and comments to authors that they might find helpful in planning their study.

Reviewer #1: Dear author,

Thank you for returning my requests. I am satisfied with the manuscript.

Congratulation for the work.

Reviewer #2: All comments have been adequately addressed.

Reviewer #3: The author adjusted all questions related to the doubts and suggestions of the reviewers. The proposed Protocol is adequate and can be replicated in other research. This study is extremely important for the population in question.

7. PLOS authors have the option to publish the peer review history of their article (what does this mean?). If published, this will include your full peer review and any attached files.

Reviewer #1: **Yes: **Valeria Feijo Martins

Reviewer #2: No

Reviewer #3: No

---

## [Editor Report · Acceptance letter]

5 Nov 2021

PONE-D-20-25040R1 

The effects of supervision on three different exercises modalities (supervised vs. home vs.  supervised+home) in older adults: Randomized controlled trial protocol 

Dear Dr. Costa:

I'm pleased to inform you that your manuscript has been deemed suitable for publication in PLOS ONE. Congratulations! Your manuscript is now with our production department. 

Kind regards, 

on behalf of

Dr. John W. Apolzan 

Academic Editor

PLOS ONE